# University Library Space Renovation Based on the User Learning Experience in Two Wuhan Universities

**DOI:** 10.3390/ijerph191610395

**Published:** 2022-08-20

**Authors:** Lei Peng, Wenli Wei, Yichen Gong, Ruiying Jia

**Affiliations:** 1School of Architecture & Urban Planning, Huazhong University of Science and Technology, Wuhan 430074, China; 2Hubei Engineering and Technology Research Center of Urbanization, Wuhan 430074, China; 3Department of Mathematics, New York University, 70 Washington, Square South, New York, NY 10012, USA

**Keywords:** university library, user experience, space renovation

## Abstract

University library spaces play an important role in the learning experience of students. However, the traditional designs for these learning spaces no longer meet the needs of users, and researchers have been turning their attention to university library space renovation. By combing existing theories and practices, this study determined a framework of six university library space renovation design principles and subsequently conducted a survey to examine university library space user learning experience in two university libraries in Wuhan, China. Data analysis was conducted using SPSS. From the questionnaire-based survey results, this study determined seven design elements that affect the learning experience of university library users. The results of binary logistic regression showed that two elements, indoor physical space comfort and indoor acoustic environment comfort, have positive effects on the frequency and length of visits to the library. Key spatial elements that can promote library space users’ learning experience were also identified, thus providing data that can reliably inform future design strategies for the space renovation of university libraries.

## 1. Introduction

The spatial renovation of university libraries has become a focus of construction worldwide since the beginning of the 21st century. Library space renovation refers to the overturning of the traditional space layout; this entails the rebuilding of the indoor and outdoor spaces and virtual spaces of libraries to meet users’ material and spiritual needs and make the space concordant with technological development, people’s cultural demands, and the transformation of the philosophy behind running libraries [1].

The user experience (UX) is the design practice of building research tools and services from a user-centered perspective. The motivation for implementing UX approaches in libraries is to actively focus on the individual needs of users, allowing libraries to design and deliver relevant services in a more targeted manner [2]. The library UX is focused on the behavioral experience of learning activities. Currently, the user learning experience in libraries is rapidly evolving toward “human interaction in service use,” which is considered an important factor in the value and development of contemporary university libraries [3].

The motivation for library space renovation stems from three main reasons. First, higher educational objectives have recently changed from cultivating purely professional talents to comprehensive and innovative talents; with this change, the simple storage and one-way transmission of knowledge in libraries no longer fits the mission of contemporary universities. Second, along with the development of constructivist learning theory in the pedagogy field, today’s higher education institutions have started emphasizing collaborative and social learning more than ever before. Third, the development of information technology has revolutionized readers’ information-seeking habits, making it important to provide relevant device technology and Internet support in the learning environment to improve users’ learning efficiency and learning experience [4]. Furthermore, students’ learning styles and demands related to the library space have also changed.

For the last two decades, there has also been much research conducted on the assessment of library space reengineering, bringing together many post-occupancy evaluations and studies involving anthropological approaches that have become a core component of the library space renovation process. The assessment of library spaces focuses on UX and is conducted based on activity types, learning preferences, space type needs, and social needs. One of the earliest space value assessments was conducted at Northeastern University in the United States. Additionally, the Tools for the Evaluation of Academic Library Spaces (TEALS) project at Deakin University in Australia made seven recommendations for space renovation. Another such assessment was conducted at the University of North Carolina at Greensboro; this evaluation project offered a progressive puzzle approach to assessing a space while renovating. There has also been such an assessment at Sheffield Hallam University, which proposed ten elements of spatial renovation based on user surveys [5].

After more than 20 years of development, the layout of the library space of universities worldwide has changed significantly. The aforementioned inventory-based approaches to space assessment have provided directional guidance for space renovation and new constructions at colleges and universities. However, the focus of many of these assessments varies, with some focusing on which space elements attract students to the library and others focusing on space utilization. Nonetheless, these two variables do not fully overlap with the student learning experience. From an architectural design perspective, the need remains to explore the specific space design elements that influence students’ learning experience.

First, this study was established on the basis of a literature review to summarize and sort out the elements of spatial renovation design in university libraries, and on this basis a research framework was conducted (Figure 1). Then we used an empirical questionnaire-based survey with samples from two university libraries in Wuhan to collect feedback on and evaluate library space performance from the perspective of the user learning experience (including four levels: physical environment, interior decoration, service facilities, and different types of space perception). It explored how architectural design can enhance and facilitate users’ learning experience in libraries, as well as identified which spatial elements impact the learning experience and which affect users’ frequency and length of visits to the library. Evidence-based strategies are provided to guide future university library space renovation endeavors.

## 2. Principles of Library Space Renovation

Currently, university libraries, as academic resource centers in universities, are undergoing a new phase of spatial transformation. Such transformation is related to the expansion of the library’s role in the campus ecological network [6]. Nowadays, users come to the library not only for learning activities but also for socializing, recreation, and other uses. Accordingly, Jamieson [7] stated that the space layout, decoration, home comfort, and interior aesthetics are important elements of the learning environment and should match the user’s needs. Zeivots and Schuck’s [4] study showed that students agreed that the physical spaces of learning environments should work together with the related virtual spaces and technological devices to support their learning activities. Moreover, Lam et al. [8], in a study on post-use evaluation of learning spaces, found that the interior design of learning spaces is an important factor that affects students’ learning experience and effectiveness. The spatial contradictions between individual and collaborative learning can be better resolved by having a diverse and flexible mix of furniture in the learning space [9]. By exploring different elements of the spatial design characteristics of the user learning experience, we have listed and organized the descriptions of various studies on the design principles that can influence the contemporary user learning space experience (Table 1 and Table 2).

## 3. Methods

### 3.1. Research Case Selection

The participant universities were Wuhan University (WHU) and Wuhan University of Technology Nanhu Campus (WUT), both in Wuhan. These two universities were conveniently selected because, first, they are both located in Wuhan, and the study authors are from Huazhong University of Science and Technology in Wuhan. Thus, the research sites were near the researchers’ location, which facilitated the research process. Second, both libraries were completed or renovated after the 2010s and are comparable in scale. WHU’s library completed its latest spatial reengineering in 2011 and has a floor area of 35,548 m^2^; WUT’s library completed its latest reengineering in 2016 and officially opened in 2018 with a floor area of 48,800 m^2^ (Table 3).

Third, both cases are representative of the current space reengineering of university libraries in China. WHU, as one of the earliest university libraries in China to build a learning sharing space, can be considered a “wind vane” of space reengineering, albeit without resulting in any overall directional change in space reengineering. As a newly built library, WUT represents the latest domestic design trends for university libraries. The types of internal learning spaces in these libraries can be divided into the reading, individual learning, group learning, and leisure learning spaces (as shown in Table 4). We aimed to analyze the difference in users’ learning experiences in university library spaces. The study was approved by the ethics committee of the Tongji Medical College of Huazhong University of Science and Technology (2022-S052), Wuhan, China. Written informed consent was obtained from the participants prior to completing the questionnaire, and oral consent was obtained for the web-based questionnaire.

### 3.2. Questionnaire Design

This was a comparative study using a quantitative method. The questionnaires were distributed in both online and offline formats. First, the researchers used social networks (WeChat group and QQ group) to disseminate the questionnaire forms made by Wen Juanxing software. Second, the researchers distributed information in the libraries, contacting library patrons and asking whether they would like to voluntarily complete the online questionnaire. The questionnaires were distributed from May to July 2022.

In the first part of the survey (show as Appendix A), users were asked to provide personal information about their sex, year of college, and major. The second part of the survey comprised a questionnaire with a scale for user satisfaction with the spatial design characteristics of the university library (responded on a five-point Likert scale ranging from 1–5 with 1, *strongly unaffected*; 2, *somewhat unaffected*; 3, *neither affected nor unaffected*; 4, *somewhat affected*; 5, *strongly affected*), which in turn contained 25 design elements. These elements and dimensions were summarized based on a review of the literature on university library space design principles, studies related to learning spaces, and the research by Freeman et al. [26], Gayton [28], and Holder and Lange [29]. The final set of evaluation factors was extracted by combining the ideas of “users’ needs for diverse learning spaces” proposed in the aforementioned literature.

## 4. Results

### 4.1. Descriptive Statistics

For data analysis of the questionnaire data, SPSS version 26.0 was used. In total, 438 questionnaires were distributed and 435 were collected. After eliminating three repeated and four invalid responses, 428 valid responses were finally used for analysis, including 211 from WHU and 217 from WUT. The effective recovery rate of the questionnaires was 97%.

Through a reliability analysis of the university library space user experience questionnaire, the Cronbach’s alpha value was 0.950 (Table 5), indicating excellent reliability. After conducting rounds of discussions with experts in the field, we deleted some items based on these discussions; then, we checked whether the Cronbach’s alpha value of the scale changed after the deletion. This process informed the decision-making regarding which items to delete. Finally, three items were deleted (Table 6). The remaining 39 items were subject to subsequent factor analysis.

Descriptive statistics of the sample are shown in Table 7. In total, 54.2% were men, and 45.8% were women, with a relatively even gender ratio of users in the study sample. Regarding educational level distribution, 46.5% of the total library users in WHU and WUT are undergraduates, and 51.7% are master’s and doctoral students. In general, the users are mainly undergraduate students. Regarding majors, the highest percentages appeared for science, engineering, agriculture, and medicine.

As shown in Table 8, regarding the frequency of visits to the library, users who come to the library every week accounted for 46.0%, followed by users who come every month (24.5%). Regarding the distribution of length of visits to the library, the number of users who spent more than three hours each time they went to the library was the highest, accounting for 46.0%, followed by the number of students who came within 1–3 h, accounting for 36.0%. Regarding purposes for visiting the library (as shown in Table 9), users who come to the library for studying alone accounted for the highest proportion, at 40.5%.

### 4.2. Factor Analysis of the Questionnaire and Data Analysis of Survey Responses

Factor analysis was performed to reduce the large amount of data collected from the questionnaires into smaller, manageable sets of components (potentially clustered factor groups) for analysis and discussion. Factors were extracted using principal component analysis with Varimax rotation and Kaiser–Meyer–Olkin (KMO) normalization methods using the SPSS FACTOR program. The results showed a KMO value of 0.938—indicating the achievement of excellent metrics—and a very low correlation significance level (*p*-value) of 0.000 (i.e., <0.05) in Bartlett’s spherical test (Table 10). This validates that factor analysis can be used to analyze scale data with sufficient evidence for further statistical analysis. That is, the scales are suitable for factor analysis.

As listed in Table 11, seven potential clustered factors were generated after factor extraction and rotation. The percentage of total variance explained was 65.608%; as shown in Figure 2, the seven-dimension model represented the variables of the entire questionnaire appropriately.

The results of the component matrix after rotation by factor analysis are shown in Table 12. These results were analyzed in combination with discussions with experts; specifically, we examined the changes in the cumulative explanatory variance of the remaining items after removing a specific item, then we decided whether to remove that item. We removed item by item several times until the extracted elements and the text items contained in the elements were more consistent with the theoretical architecture. Through this process, the original 39 items were reduced to 28 items. Based on the total variance explained, the original six factors as Table 2 (physical space environment; supporting facilities and services; space availability; interior decoration; Furniture type and comfort; Space layout accessibility) were reclassified into seven categories. The results of naming the seven first-level elements are shown in Table 13, including (E1) physical environment self-control; (E2) physical environment comfort; (E3) interior space aesthetics; (E4) interior acoustic environment comfort; (E5) interior space use comfort; (E6) interior space use flexibility; (E7) complete guidance and equipment.

### 4.3. Binary Logistic Regression Analysis

#### 4.3.1. Frequency of Visits to the Library

Using binary logistic regression, the seven factors obtained were used to explore the variables influencing students’ frequency and length of visits to the library. First, the independent variables were the seven factors that affect the university library user learning experience, and the dependent variable was the frequency of visits to the library. Results showed (Figure 3) that the total percentage of daily and weekly library users was 53.2%. Therefore, we dichotomized the frequency of coming to the library by daily and weekly visits (F1 = daily + weekly vs. F0 = monthly + quarterly + basically not going + other). The Hosmer and Lemeshow test results are shown in Table 14; the model showed a good fit to the data (*p* = 0.096, >0.05).

The results of the binary logistic regression analysis for the frequency of visits to the library are shown in Table 15. Specifically, the *p*-values for physical environment comfort (E2; *p* = 0.001, <0.05) and indoor acoustic comfort (E4; *p* = 0.037, <0.05) were significant and positively correlated with the frequency of visits to the library. The *p*-value of interior space aesthetics (E3; *p* = 0.001, <0.05) was also significant and negatively correlated with the frequency of visits to the library.

#### 4.3.2. Binary Logistic Regression Analysis of Length of Visits to the Library

Here, the independent variables are the seven factors that affect university library user learning experience, and the dependent variable is the length of visits to the library. The total time spent in the library for each study session was used to measure the length of visits to the library. According to the results (Figure 4), 82.0% of the students spent more than one hour in the library, so we chose one hour as the cutoff value and treated the length of visit to the library as a dichotomous variable (T1 = more than one hour/time vs. T0 = less than one hour/time; the latter was used as the reference group). The Hosmer and Lemeshow results are shown in Table 16 and indicate that the model fits the data well (*p* = 0.747, >0.05).

The results of the binary logistic regression analysis of the length of time visiting the library are shown in Table 17. The *p*-values of physical environment comfort (E2; *p* = 0.032, <0.050) and indoor acoustic comfort (E4; *p* = 0.001, <0.050) were significant and positively correlated with the length of visit to the library. The *p*-value of interior space aesthetics (E3; *p* = 0.003, <0.05) was significant and negatively correlated with the length of visit to the library.

## 5. Discussion

### 5.1. Major Findings

#### 5.1.1. Seven Spatial Factors Affecting User Experience

This study conducted a factor analysis based on a questionnaire scale to obtain seven spatial factors that affect students’ learning experience in the library: (E1) physical environment self-control; (E2) physical environment comfort; (E3) interior space aesthetics; (E4) interior acoustic environment comfort; (E5) interior space use comfort; (E6) interior space use flexibility; and (E7) complete guidance and equipment. The contribution of these spatial factors is shown in Figure 5. A binary logistic regression analysis was then performed to relate each of these seven factors to users’ frequency and length of visit to the library to further explore the spatial factors that enhance these two variables.

E1: Physical environment self-control

In one empirical study, Beckers et al. [30] found users strongly needed personal control over noise, temperature, and room illumination in the learning space. Another study showed that differences in the quantity and quality of light can have different effects on the comfort level of the learning environment [31]. Hence, a room with natural lighting, due to its uncontrollability, makes it more urgent for users to have personal control and the ability to adjust artificial light sources.

Temperatures above the comfort zone (25 °C) can significantly negatively impact learners’ learning performance [15], and changes in outdoor temperatures during different seasons require learners to have the autonomy to adjust artificial temperatures according to their needs in order to arrive at a comfortable temperature zone for learning.

Flexible and movable furniture can be used to meet the changing needs of users’ learning or social activities and enhance the user’s sense of domain over the learning space, thus creating an overall atmosphere of comfort [9]. The lightness and mobility of the furniture also promote flexibility in the use of the space, allowing learners to quickly and autonomously modify the layout of the learning space to meet the needs of different learning activities and improve their space use experience.

E2: Physical environment comfort

For users, a learning space that provides continuous and prolonged open services enhances their learning experience [5]. When natural lighting is insufficient, adequate artificial lighting can be used to meet the learners’ lighting needs. By contrast, when the natural lighting is adequate, it provides a comfortable lighting environment. An appropriate number of power outlets can also extend the length of time learners spend using electronic devices and enhance their space learning experience. Good ventilation in the room can improve air quality, which is positive for learners’ health and concentration [13]. Providing sufficient storage space enables learners to have a place where they can store their belongings, which improves the orderliness of the learning space and facilitates the learning experience. An adequate supply of tables and chairs can increase the number of learners coming to the library and provide learners with the freedom to use the tables and chairs at their will. The 21st century learning environment is closely linked to wireless broadband (Wi-Fi) networks and mobile communication devices [18]. Accordingly, a library with high-quality Wi-Fi provides flexibility and choice for learners to engage in different learning activities.

E3: Aesthetics of indoor space

The indoor greenery arrangement can have a soothing effect on the spirits of space users [23]. Different colors have different effects on the psychological perception of learners, and appropriate interior coloring can promote learners’ positive learning emotions [32]. A beautiful view through a window near a study seat can also have a positive impact on students’ physiological and psychological states. Furthermore, a well-designed interior creates a relaxed and comfortable learning environment and improves the learning experience for learners. Again, adequate natural lighting ensures that learners have a comfortable light environment.

E4: Indoor acoustic comfort

Background noise can weaken learners’ concentration [11]. A quiet acoustic environment can thus reduce the external environment’s interference in the learning process of library users. Furthermore, having access to an acoustic environment that provides privacy to its users can motivate the engagement in different learning activities, such as group discussions; this is because, in such environments, users need not worry about the sound from their learning activities disturbing other learners.

E5: Indoor space comfort

Providing learners with various study spaces in the library enables meeting different needs and increasing their frequency of visits. Furthermore, a comprehensive range of services and facilities associated with the library provides learners with a better library learning experience [4].

E6: Flexibility in the use of interior space

Library space design is gradually moving toward the trend of being “rearrangeable” [27]. In other words, the space layout should be flexible and changeable to meet the needs of learners. Users can also perceive that the learning behavior of others promotes learning at the individual level.

E7: Complete guidance and equipment

With the development of Internet technology, library users have started to increase their demands for computers, scanners, printers, and copiers [33]. Well-equipped facilities can improve the quality of library services and support more learning activities for students. A clear guidance system in the library can also improve the legibility and accessibility of the space and improve the learners’ perceptions of space use.

#### 5.1.2. Influence of Spatial Factors on Frequency of Visits and Length of Time in the Library

Our findings show that, first, physical environment comfort (E2) and interior acoustic environment comfort (E4) promote the frequency of student visits to the library. This result reaffirms that enhancing the physical elements of the learning environment, such as light and appropriate background sound, can enhance the space’s attraction for students [5].

Second, physical and indoor acoustic comfort can encourage students to spend longer periods in the library to study or engage in other activities. As with the results of frequency of visits, the lighting [12], temperature [15], and ventilation [13] of the space may also encourage users to spend more time in the library once they visit it. Meeting the needs of students in the library for adequate power outlets and good Wi-Fi signal [34] can extend the length of time users study in the library. Furthermore, we confirm Harrop and Turpin’s [5] study of student demand for open hours: if the library has longer and continuous opening hours, it may be able to meet the needs of a wider range of users for longer periods. Less background noise can also improve the concentration of users in the library.

Our findings also showed that the aesthetics of the interior space negatively affect the frequency and length of visits to the library. These results are inconsistent with the findings of prior studies, which have shown that elaborate interior decoration [35,36] and beautiful overall colors [32] positively affect learners. The reasons for this may be because first, sophisticated decoration may attract more learners to the library, and excessive numbers of people create noise and other distractions that can affect individual learning experience [37]. Second, the two Chinese university libraries included in this study did not use a diverse range of colors, implying that learners may lack experience in the relationship between color and the learning experience. Third, the artificial lighting in Chinese libraries tends to be continuous and sufficient. Natural light that is too direct may cause glare and disrupt the learning experience. Fourth, indoor greenery may attract insects and emit odors, affecting students’ learning experience. Fifth, the changing landscape outside the window may interfere with the learning state of indoor users.

### 5.2. Strengths and Limitations

One of the strengths of this study is that it was conducted in two representative university libraries in Wuhan. Our findings support some conclusions from previous literature that physical space comfort can attract learners to the library [12] and indoor acoustic comfort can promote a better learning experience [10]. Furthermore, space layout accessibility and longer hours of operation can encourage more frequent visits to the library. These findings enrich the literature by providing suggestions as to what should be prioritized in library space reengineering endeavors.

We also adopted the user learning experience as a new entry point for assessing the value of library learning spaces. Using this approach, this study quantitatively demonstrates that the Internet and electronic devices play an important role in users’ learning experience in the 21st century, as well as that the location of the library on the campus can influence the frequency of visits to the library.

This research also has some limitations. First, the questionnaires were primarily administered to university students, so the opinions and comments of faculty and librarians were not collected. Second, the study was limited to the main libraries of two universities, so further research is needed to investigate more related settings. These future studies will allow us to have a more accurate assessment of the impact of variables related to library space on the user learning experience.

## 6. Conclusions

Through empirical research, we have appropriate evidence to identify the elements of university library space transformation that impact the user learning experience (Figure 6). This quantitative study determined seven first-level design elements of university library spaces that may influence user learning experience. Results showed that physical environment comfort (E2) and interior acoustic environment comfort (E4) of university libraries were positively correlated with the frequency and length of visits to the library. Therefore, in the process of space reengineering for university libraries, the following two suggestions are made: (1) improve the comfort of the physical space and (2) improve the indoor acoustic environment of the library.

## Figures and Tables

**Figure 1 ijerph-19-10395-f001:**
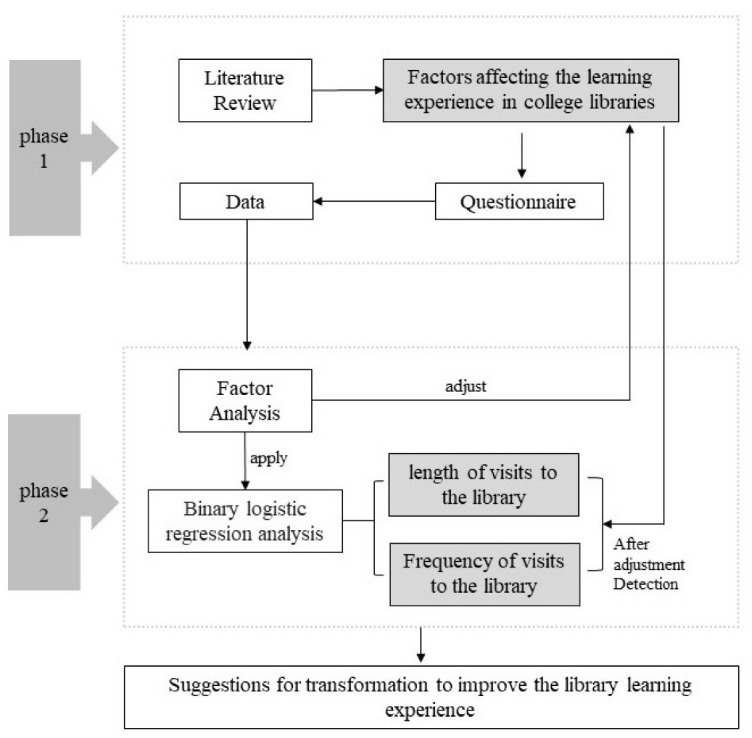
Research framework.

**Figure 2 ijerph-19-10395-f002:**
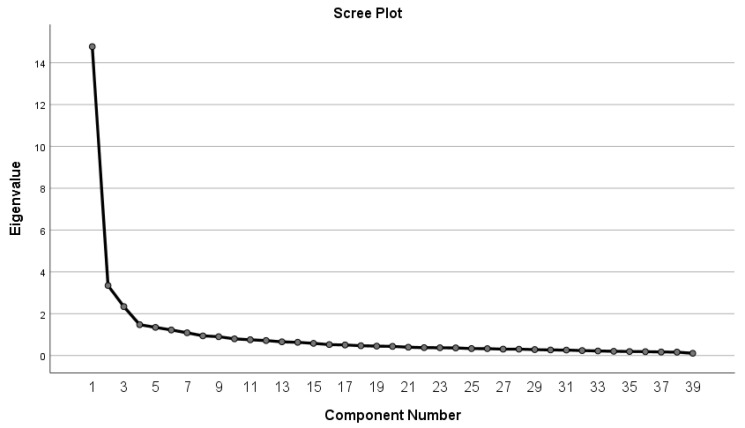
Scree Plot.

**Figure 3 ijerph-19-10395-f003:**
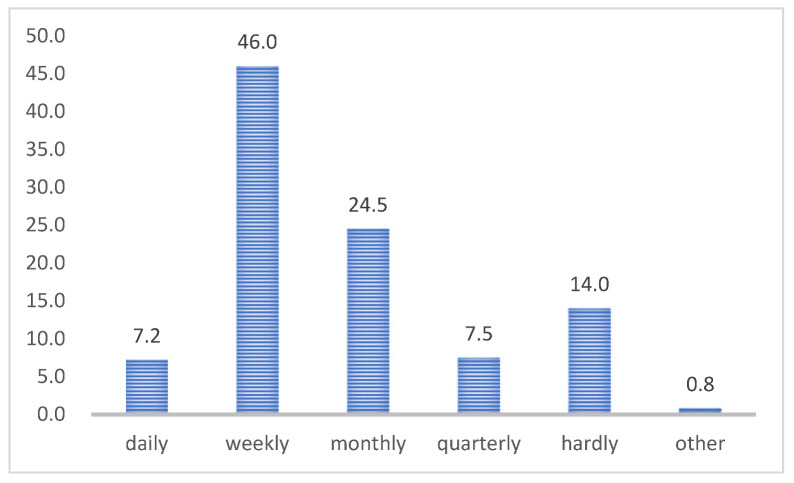
Visits’ frequency of sample distribution.

**Figure 4 ijerph-19-10395-f004:**
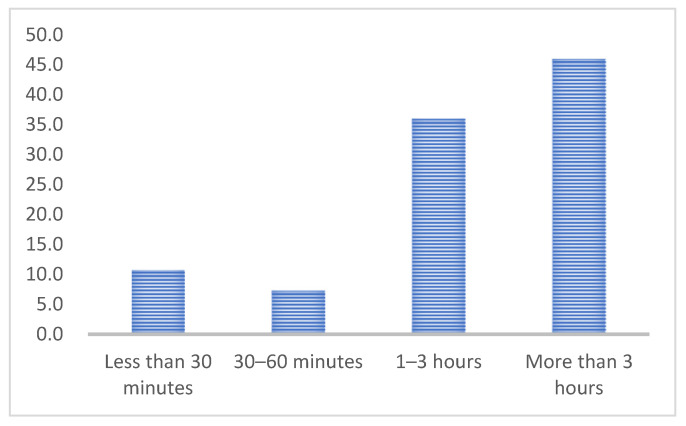
Visits’ length of sample distribution.

**Figure 5 ijerph-19-10395-f005:**
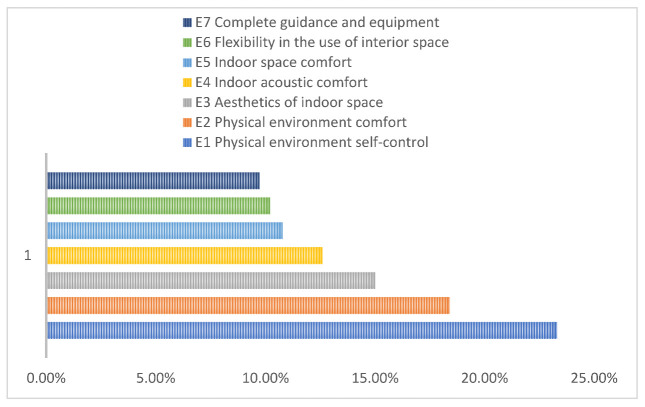
Contribution of spatial factors.

**Figure 6 ijerph-19-10395-f006:**
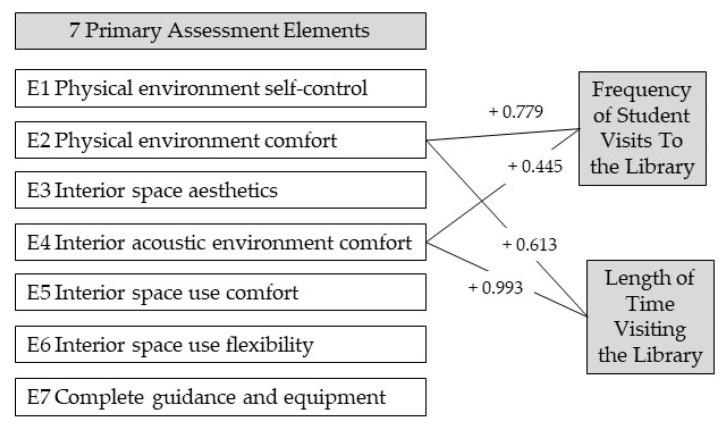
Empirical research conclusion.

**Table 1 ijerph-19-10395-t001:** Spatial design features of learning spaces that affect user experience based on available literature.

Space Design Features	Sources
Sound; lighting (natural, artificial light); ventilation; temperature	[5,10,11,12,13,14,15]
Storage; power and outlet configurations; network and Wi-Fi quality; electronic equipment configurations such as printers, copiers, etc.; food availability; open hours; directional signage	[4,5,16,17,18,19]
Diversity of space types; individual study spaces; group discussion rooms	[5,20]
Decorative sophistication; colors; indoor greenery	[8,21,22,23]
Diversity of furniture types; home comfort; furniture flexibility	[24,25,26]
The geographically centralized location of the library	[27]

**Table 2 ijerph-19-10395-t002:** Checklist for assessing the university library space user experience.

Primary Assessment Element	Secondary Assessment Element
a. Physical space environment	Interior acoustic environment
	Interior lighting status (natural/artificial lighting)
	Interior ventilation status
	Interior temperature status
b. Supporting facilities and services	Storage space
	Amount of power supply and outlet configuration
	Quality of the network and Wi-Fi signal in the building
	Configuration of electronic equipment such as printers and copiers
	Light refreshment supply
	Library opening hours
	Guiding signs
c. Space availability	Supply of tables and chairs for study and reading rooms
	Supply of individual study rooms
	Supply of group seminar rooms
	Supply of open, collaborative spaces
	Supply of leisure study spaces
	Supply of diversified learning spaces
d. Interior decoration	Interior decoration refinement
	Interior decoration color aesthetics
	Interior greenery configuration
e. Furniture type and comfort	Diversification of furniture types
	Furniture comfort
	Furniture flexibility
f. Space layout accessibility	Nearby residence
	Library cafeteria

**Table 3 ijerph-19-10395-t003:** Basic information about the libraries of the two universities.

School Name	Basic Information			
	Site	Total Floor Area	Completion Date	Date of Expansion/Construction
Wuhan University	Core area of the campus teaching area	35,548	1981	2014
Wuhan University of Technology	West of the teaching	48,800	2016	/

**Table 4 ijerph-19-10395-t004:** Distribution of learning space types in the two universities.

School Name	Partial Learning Space Distribution	Type of Learning Space
Reading or Study	Personal	Group	Leisure
Wuhan University	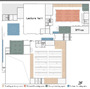	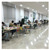	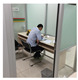	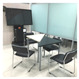	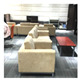
Wuhan University of Technology	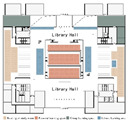	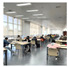	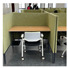	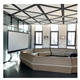	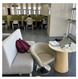

**Table 5 ijerph-19-10395-t005:** Reliability statistics for the university library space user experience questionnaire.

Cronbach’s Alpha	Cronbach’s Alpha Based on Standardized Items	Number of Items
0.950	0.954	42

**Table 6 ijerph-19-10395-t006:** Results for deleted items of the university library space user experience questionnaire.

Item-Total Statistics				
	Scale Mean If Item Deleted	Scale Variance If Item Deleted	Corrected Item-Total Correlation	Cronbach’s Alpha If Item Deleted
Question 7.3. Library adjacent to dormitory and cafeteria	163.46	546.305	0.354	0.950
Question 12.4. Personal study space is visible from the outside line of sight	164.00	543.909	0.381	0.950
Question 13.4. Group study space is visible from the outside line of sight	163.93	543.267	0.379	0.950

**Table 7 ijerph-19-10395-t007:** Descriptive statistics for all participants (*n* = 428, Wuhan, China, 2021–2022).

Variable	Option	Frequency	Percent
Gender	Male	232	54.2
	Female	196	45.8
Degree	Undergraduate	199	46.5
	Master’s	186	43.5
	Doctorate	35	8.2
	Faculty and Researchers	7	1.6
	Other	1	0.2
Major	Philosophy, economics, law	49	11.4
	Pedagogy, literature, history	84	19.6
	Science, engineering, agriculture, medicine	246	57.5
	Military science, management science, science of art	49	11.4

**Table 8 ijerph-19-10395-t008:** Descriptive statistics of frequency and length of visits to the library.

Variable	Option	Frequency	Percent
Frequency of visits to the library	Daily	31	7.2
	Weekly	197	46.0
	Monthly	105	24.5
	Quarterly	32	7.5
	Hardly	60	14.0
	Others	3	0.7
Length of visits to the library	Less than 30 min	46	10.7
	30–60 min	31	7.2
	1–3 h	154	36.0
	More than 3 h	197	46.0

**Table 9 ijerph-19-10395-t009:** Descriptive statistics of the purpose for visiting the library.

Variable	Option	Frequency	Percent	Percentage of Cases
The purpose of visiting the library	Study alone	364	40.5%	85.0%
	Complete group work	106	11.8%	24.8%
	Borrow and return books	187	20.8%	43.7%
	Endorsement, speech practice	40	4.4%	9.3%
	Use electronic resources in the library	114	12.7%	26.6%
	Communication, interaction, and other social activities	58	6.5%	13.6%
	Entertainment (watching movies, listening to music, playing games, etc.)	18	2.0%	4.2%
	Others	12	1.3%	2.8%
Total		899	100.0%	210.0%

**Table 10 ijerph-19-10395-t010:** Results of the Kaiser-Meyer–Olkin and Bartlett’s Test.

KMO and Bartlett’s Test		
Kaiser–Meyer–Olkin measure of sampling adequacy		0.938
Bartlett’s test of sphericity	Approximate chi-square value	10,996.908
	Degree of freedom (df)	741
	Significance level (*p*-value)	0.000

**Table 11 ijerph-19-10395-t011:** Exploratory factor analysis results for the university library space user experience questionnaire: total variance results.

Total Variance Explained									
Component	Initial Eigenvalues			Extraction Sums of Squared Loadings			Rotation Sums of Squared Loadings		
	Total	% of Variance	Cumulative %	Total	% of Variance	Cumulative %	Total	% of Variance	Cumulative %
1	14.771	37.875	37.875	14.771	37.875	37.875	5.963	15.289	15.289
2	3.348	8.584	46.459	3.348	8.584	46.459	4.708	12.073	27.362
3	2.340	5.999	52.457	2.340	5.999	52.457	3.840	9.846	37.207
4	1.474	3.779	56.236	1.474	3.779	56.236	3.221	8.260	45.468
5	1.346	3.451	59.687	1.346	3.451	59.687	2.756	7.065	52.533
6	1.222	3.134	62.821	1.222	3.134	62.821	2.609	6.691	59.224
7	1.087	2.788	65.608	1.087	2.788	65.608	2.490	6.385	65.608

**Table 12 ijerph-19-10395-t012:** Exploratory factor analysis for the university library space user experience questionnaire: rotated component matrix.

Rotated Component Matrix							
	Component						
	1	2	3	4	5	6	7
Q13.1	0.854						
Q12.1	0.833						
Q12.2	0.778						
Q13.2	0.764						
Q14.1	0.654						
Q14.2	0.527						
Q11.1	0.523						
Q14.3	0.517						
Q7.7		0.701					
Q7.3		0.659					
Q8.2		0.656					
Q10.3		0.622					
Q8.4		0.621					
Q8.3		0.568					
Q10.1		0.547					
Q7.5		0.520					
Q10.2		0.518					
Q7.1							
Q9.3			0.812				
Q9.2			0.745				
Q8.5			0.731				
Q9.1			0.707				
Q8.1			0.509				
Q11.4							
Q12.3				0.660			
Q13.3				0.553			
Q11.3				0.530			
Q11.2				0.515			
Q14.4							
Q7.6					0.707		
Q7.4					0.596		
Q7.2					0.561		
Q10.7							
Q13.5						0.753	
Q13.4						0.721	
Q11.5						0.561	
Q10.5							0.775
Q10.6							0.697
Q10.4							

**Table 13 ijerph-19-10395-t013:** Spatial factors that influence the university library space user experience.

Primary Assessment Element	Secondary Assessment Element
Physical environment self-control	Adjustable light level
	Adjustable temperature level
	Casual and comfortable furniture
	Furniture is lightweight, flexible, and movable
	Sight lines
	Space provides opportunities for social interaction
Physical environment comfort	Good learning atmosphere
	Library has continuous and long opening hours
	Sufficient artificial lighting in the library
	Sufficient power outlets in the library
	Temperature inside the library is suitable (warm in winter and cool in summer)
	Library has good ventilation
	Adequate storage space in the library
	Adequate supply of tables and chairs
	Good quality Wi-Fi signal in the building
Interior space aesthetics	Interior greenery is properly arranged
	Beautiful overall interior coloring
	Beautiful view outside the window
	Overall interior decoration is exquisite
	Ample natural light inside the building
Interior acoustic environment comfort	Acoustic environment is sufficiently private
	Low background noise
Interior space use comfort	Various types of learning spaces
	Abundant service facilities
Interior space use flexibility	Layout of the space can be flexibly changed
	Perceive the learning behavior of others
Complete guidance and equipment	Computer workstations in the pavilion
	Clear signage in the pavilion

**Table 14 ijerph-19-10395-t014:** Hosmer and Lemeshow test results.

Hosmer and Lemeshow Test			
Step	Chi-Square	df	Sig.
1	13.491	8	0.096

**Table 15 ijerph-19-10395-t015:** Results of logistic regression analysis for frequency of visits to the library.

							95% CI for Exp (B)	
	B	S.E.	Wald	df	Sig.	Exp (B)	Lower	Upper
E1	−0.230	0.242	0.900	1	0.343	0.795	0.494	1.278
**E2**	**0.779**	0.226	11.868	1	**0.001**	2.179	1.399	3.393
**E3**	**−0.520**	0.164	10.093	1	**0.001**	0.594	0.431	0.819
**E4**	**0.445**	0.213	4.363	1	**0.037**	1.560	1.028	2.368
E5	−0.012	0.146	0.007	1	0.932	0.988	0.743	1.314
E6	−0.115	0.152	0.571	1	0.450	0.891	0.662	1.201
E7	0.020	0.116	0.029	1	0.864	1.020	0.813	1.280
Constant	−1.795	0.790	5.163	1	0.023	0.166		

Note. E1: physical environment self-control; E2: physical environment comfort; E3: interior space aesthetics; E4: interior acoustic environment comfort; E5: interior space use comfort; E6: interior space use flexibility; E7: complete guidance and equipment.

**Table 16 ijerph-19-10395-t016:** Hosmer and Lemeshow test results.

Hosmer and Lemeshow Test			
Step	Chi-Square	df	Sig.
1	5.096	8	0.747

**Table 17 ijerph-19-10395-t017:** Results of logistic regression analysis of length of visit to the library.

							95% CI for Exp (B)	
	B	S.E.	Wald	df	Sig.	Exp (B)	Lower	Upper
E1	0.174	0.339	0.263	1	0.608	1.190	0.612	2.314
**E2**	**0.613**	0.286	4.595	1	**0.032**	1.846	1.054	3.233
**E3**	**−0.702**	0.237	8.777	1	**0.003**	0.496	0.312	0.789
**E4**	**0.993**	0.291	11.667	1	**0.001**	2.700	1.527	4.773
E5	0.093	0.185	0.250	1	0.617	1.097	0.763	1.576
E6	−0.395	0.232	2.897	1	0.089	0.674	0.427	1.062
E7	−0.295	0.179	2.714	1	0.099	0.745	0.525	1.058
Constant	−1.244	0.818	2.314	1	0.128	0.288		

Note. E1: physical environment self-control; E2: physical environment comfort; E3: interior space aesthetics; E4: interior acoustic environment comfort; E5: interior space use comfort; E6: interior space use flexibility; E7: complete guidance and equipment.

## Data Availability

Not applicable.

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
