# Peer review of "University Library Space Renovation Based on the User Learning Experience in Two Wuhan Universities"

_ijerph, 2022, doi:10.3390/ijerph191610395_

Round 1

Reviewer 1 Report

Peng et al. presented a detailed study on library settings in two universities situated in Wuhan. 

The finding is based on the survey conducted on these two campuses. The data is compiled using SPSS software and performed some statistical analysis.

The main negative point of this version is the presentation of contents. the reader may lost in between reading the result section. They should consider putting some plots/tables in the supplementary files (*if the journal supports it).   Also, the conclusion section must include the main findings and other information in crispy format.

Some other comments:

>Mention the keywords. (If authors had intended to put).

> Change the sentence. here 4 is confusing. It seems like a reference number! If yes, then please check some published articles and cite them accordingly.

> The response points could be written as: very affected->-> strongly affected. Strongly can be a suitable choice.

> Line No 342: Recheck cited reference.

Author Response

We have added some charts (Figures 1-6 and Table 3-4) that can make the article clearer especially in conclusion section

>Keywords already added to the text:university library; user experience; space renovation

>This is an unnecessary symbol and has been removed

>We have revised in the text and appendix A

>We have modified the format of these two references

Reviewer 2 Report

The authors propose an interesting article to contribute to the improvement in the design of university libraries. However, I suggest major changes to the structure of the article prior to publication.

Line 9: The authors affirm that they have contributed equally to the work developed. However, this statement should appear only in the specific section on the contribution of the authors, and not after the affiliations.

Line 23: Keywords are missing

Lines 86, 88, 90, 92, 96, 142, 143. Please, review the format of the reference (also in the rest of the manuscript)

Lines 202-207. The authors say, Based on the total variance explained, the original six factors were reclassified into seven categories. The results of naming the seven first-level elements are shown in Table 12, including (E1) physical environment self-control; (E2) physical environment comfort; (E3) interior space aesthetics; (E4) indoor acoustic environment comfort; (E5) interior space use comfort; (E6) interior space use flexibility; (E7) complete guidance and equipment. Which were the original six factors? Did you mention/describe them before in the manuscript? Which is the result of the analysis that supports the decision defining 7 categories?

Lines 146-207. In the results section, you include some tables, in which it is quite difficult for the reader to realize the relationship between variables. Could you please add some graphs that support your results? (Scree plot, loading plot) The scree plot, for example, can help you to explain why you select 6 factors instead of another amount.

Lines 241-242. In this part of the manuscript, the authors relate each category with other research articles. However, in my opinion, this part of the manuscript should not be in the results section, but rather in the discussion section, as it is not describing the findings of the current research, but the relation with other research. Instead, I suggest that a graphic analysis of the division of the factors and categories, and their contribution to the model be added.

Lines 457-461. You use three pages to define two tables of the manuscript. I suggest the tables be changed to reduce their protagonism in the manuscript. For example, the scales are the same for each table, so they could be included in the question itself, as possible answers.

Author Response

Line 9: We have put this quote in the author contribution

Line 23: Keywords already added to the text:university library; user experience; space renovation

Lines 86, 88, 90, 92, 96, 142, 143: We have checked and changed the format of all references you proposed

Lines 202-207: The initial six factors are summarized in Table 2, based on the literature review (Table 1). supports the decision defining 7 categories mentioned in the second paragraph of Section 4.2, seven potential clustered factors were generated after factor extraction and rotation. The percentage of total variance explained was 65.608%; As shown in Figure 2, the seven-dimension model represented the variables of the entire questionnaire appropriately.

Lines 146-207: We have added a scree plot in chapter 4.2 (Figure 2) and a plot of the sample distribution of frequency of visit and length of stay in chapter 4.3 (Figures 3 and 4) so that the reader can see the relationship between the variables more clearly.

Lines 241-242: We have put this section in the discussion section and added a graphical analysis of the contribution of spatial factors (as in Figure 5)

Lines 457-461: We have merged the two tables and reduced the number of pages this table takes up in the article

Round 2

Reviewer 1 Report

The authors have included the suggestions and responded to my queries properly. So, the revised manuscript can be considered for the next stage process.

Reviewer 2 Report

The authors have satisfactory changed the manuscript, so I recommend the paper to be published in its current form